# Asymmetric structure of the native *Rhodobacter sphaeroides* dimeric LH1–RC complex

Kazutoshi Tani [1,8,9✉], Ryo Kanno[2,8], Riku Kikuchi[3], Saki Kawamura[3], Kenji V. P. Nagashima[4], Malgorzata Hall[2], Ai Takahashi[2], Long-Jiang Yu [5], Yukihiro Kimura [6], Michael T. Madigan[7], Akira Mizoguchi[1], Bruno M. Humbel[2] & Zheng-Yu Wang-Otomo [3,9✉]

*Rhodobacter sphaeroides* is a model organism in bacterial photosynthesis, and its light-harvesting-reaction center (LH1–RC) complex contains both dimeric and monomeric forms. Here we present cryo-EM structures of the native LH1–RC dimer and an LH1–RC monomer lacking protein-U (ΔU). The native dimer reveals several asymmetric features including the arrangement of its two monomeric components, the structural integrity of protein-U, the overall organization of LH1, and rigidities of the proteins and pigments. PufX plays a critical role in connecting the two monomers in a dimer, with one PufX interacting at its N-terminus with another PufX and an LH1 β-polypeptide in the other monomer. One protein-U was only partially resolved in the dimeric structure, signaling different degrees of disorder in the two monomers. The ΔU LH1–RC monomer was half-moon-shaped and contained 11 α- and 10 β-polypeptides, indicating a critical role for protein-U in controlling the number of αβ-subunits required for dimer assembly and stabilization. These features are discussed in relation to membrane topology and an assembly model proposed for the native dimeric complex.

[1] Graduate School of Medicine, Mie University, Tsu 514-8507, Japan. [2] Imaging Section, Research Support Division, Okinawa Institute of Science and Technology Graduate University (OIST), 1919-1, Tancha, Onna-son, Kunigami-gun, Okinawa 904-0495, Japan. [3] Faculty of Science, Ibaraki University, Mito 310-8512, Japan. [4] Research Institute for Integrated Science, Kanagawa University, 2946 Tsuchiya, Hiratsuka, Kanagawa 259-1293, Japan. [5] Photosynthesis Research Center, Key Laboratory of Photobiology, Institute of Botany, Chinese Academy of Sciences, Beijing 100093, China. [6] Department of Agrobioscience, Graduate School of Agriculture, Kobe University, Nada, Kobe 657-8501, Japan. [7] School of Biological Sciences, Department of Microbiology, Southern Illinois University, Carbondale, IL 62901, USA. [8] These authors contributed equally: Kazutoshi Tani, Ryo Kanno. [9] These authors jointly supervised this work: Kazutoshi Tani, Zheng-Yu Wang-Otomo. ✉email: ktani@doc.medic.mie-u.ac.jp; wang@ml.ibaraki.ac.jp

*R*hodobacter (*Rba.*) *sphaeroides* is the most thoroughly investigated purple phototrophic bacterium and is widely used as a model organism. The combination of its metabolic diversity, ease of growth, and well-established genetics has revealed the details of many fundamental mechanisms of photosynthesis, including photochemistry, metabolism, and regulation. The core light-harvesting (LH1) system of *Rba. sphaeroides* is characterized by the coexistence of dimeric and monomeric forms corresponding to the S- and C-shaped structures, respectively, observed in native membranes[1,2]. Both dimeric and monomeric *Rba. sphaeroides* LH1 complexes can be separately purified in a reaction-center (RC)-associated form. The so-called core complex (LH1–RC) contains a protein, PufX[3,4], whose roles in regulation of membrane morphology, core photocomplex organization, and cyclic electron transfer, have been intensively investigated[5] but still unsettled.

In a recent study, we determined the structure of a monomeric LH1–RC from *Rba. sphaeroides* strain IL106[6]. The LH1 complex forms a C-shaped structure composed of 14 αβ-polypeptides around the RC with a large ring opening. In addition to PufX, a previously unrecognized integral membrane protein, referred to as protein-U, was identified in the core complex. Protein-U is located opposite PufX on the LH1-ring opening and has a U-shaped conformation. Deletion of protein-U resulted in a mutant strain that expressed a much-reduced amount of dimeric LH1–RC, implying key roles for protein-U in stabilization of the dimeric form of LH1–RC complex[6]. A similar structure was also reported for a monomeric core complex derived from a genetically modified strain of *Rba. sphaeroides*[7].

Much effort has been devoted to determining the structure of the dimeric LH1–RC from *Rba. sphaeroides*[7–11]. Recently, a two-fold symmetric structure of the dimeric LH1–RC was reported for an LH2-deficient mutant strain of *Rba. sphaeroides* (strain DBCΩG)[7]. Dimeric LH1–RC complexes were also observed in *Rba. blasticus*[12] and *Rhodobaca bogoriensis*[13], and were suggested to exist in a few other purple bacteria[14].

Here, we present cryo-EM structures of the dimeric core complex from native *Rba. sphaeroides* IL106 and the monomeric LH1–RC from a protein-U-deleted mutant strain (strain ΔU) of *Rba. sphaeroides* IL106[15]. In contrast to the symmetric structure reported from the LH2-deficient mutant strain[7], the native LH1–RC dimer reveals an asymmetric S-shaped structure. Based on our structural information, the functions of PufX and protein-U in the *Rba. sphaeroides* LH1–RC core complex have been clarified and are discussed in the context of a model that describes the assembly and stabilization of the mature LH1–RC complex as it exists in wild-type cells.

## Results

### Asymmetric features of the native *Rba. sphaeroides* LH1–RC dimer.

The cryo-EM structure of the dimeric LH1–RC complex from native *Rba. sphaeroides* IL106 was determined at 2.75 Å resolution with *C*1 symmetry (Fig. 1, Supplementary Table 1 and Supplementary Figs. 1–6). The two halves in the LH1–RC dimer, designated as monomer A and monomer B, are not symmetric. While two PufX polypeptides and all LH1 and RC polypeptides were fully identified in the dimer, protein-U in monomer B was only partially detected, presumably due to its large disorder. No other proteins were detected in the native dimer structure.

Monomer A is rotated approximately 8˚ anticlockwise relative to the mirror image of monomer B, as determined from the dihedral angle of cross sections in the two monomers that are perpendicular to the presumed membrane surface (Fig. 1a). This rotation results in a significantly larger opening in the LH1 ring of monomer A than the LH1 ring of monomer B. The two

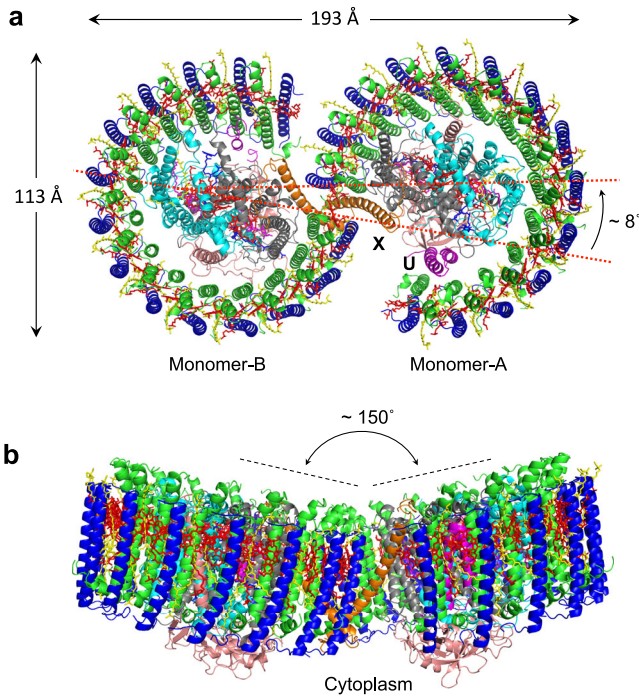

**Fig. 1 Structure overview of the dimeric LH1–RC complex from *Rba. sphaeroides* IL106. a** Top view from the periplasmic side of the membrane with PufX (orange) and protein-U (magenta) in monomer A indicated by their chain ID: X and U, respectively. **b** Side view of the core complex parallel to the membrane plane with the H-subunits (salmon) at the bottom. Color scheme: LH1-α, green; LH1-β, blue; RC-L, black; RC-M, cyan; BChl *a*, red sticks; spheroidenes, yellow sticks; BPhe *a*, magenta sticks.

monomers in the dimeric LH1–RC are arranged in a bent configuration with an angle of approximately 150° inclined toward each other of their presumed periplasmic surfaces (Fig. 1b). This topology is similar to that of a dimeric LH1–RC structure from the LH2-deficient *Rba. sphaeroides* mutant strain DBCΩG[7,11], and is consistent with the curvature of vesicle-type intracytoplasmic membrane (ICM) surface as observed in phototrophically grown native *Rba. sphaeroides* cells[16].

The *Rba. sphaeroides* LH1 dimeric complex is composed of 28 pairs of αβ-polypeptides, 56 bacteriochlorophylls (BChl) *a*, and 52 spheroidenes surrounding two RCs with an S-shaped structure. BChl-*a* molecules in the *Rba. sphaeroides* dimeric LH1 form a partially overlapping S-shaped ring with average Mg–Mg distances of 9.5 Å within an αβ-subunit and 8.3 Å between the αβ-subunits (Fig. 2a). These values are similar to those observed for the monomeric *Rba. sphaeroides* LH1 (Supplementary Table 2)[6]. The $Q_A$ site of quinones in the RC is located close to PufX, whereas the $Q_B$ site is on the side of protein-U. As in the case of monomeric *Rba. sphaeroides* LH1–RC[6], two groups of all-*trans*-spheroidenes with distinct configurations were detected in the dimeric LH1 (Fig. 2b). Again, asymmetric features were observed for the *B*-factor (an indicator of the degree of disorder) of these carotenoids in LH1 (Fig. 2b) as follows: (i) the carotenoids in monomer A have larger *B*-factors than those at corresponding positions in monomer B, (ii) group-B carotenoids (defined as those protruding on the periplasmic surface)[6] have larger *B*-factors than those of the group-A carotenoids (defined as those embedded deeply in the transmembrane region) in an αβ-subunit; and (iii) the carotenoids around the dimer junction have smaller *B*-factors than do those of the carotenoids near the ends of the LH1 ring. These results indicate

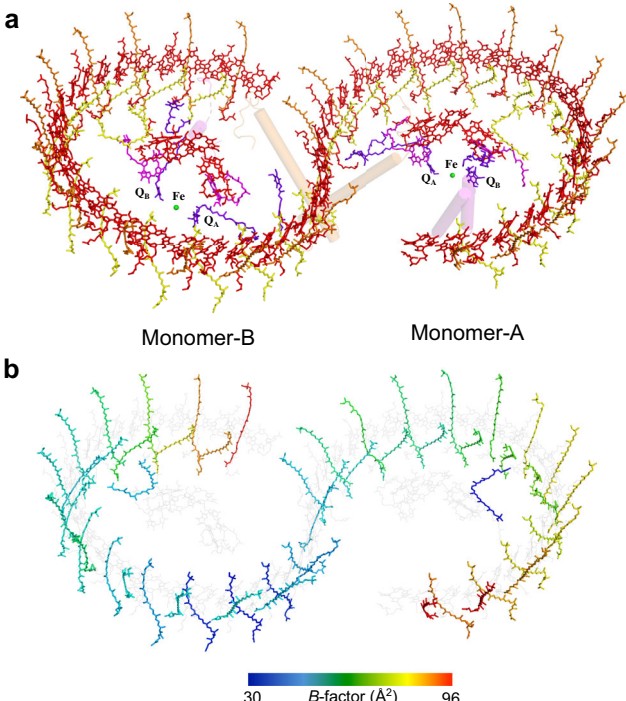

**Fig. 2 Arrangement of the cofactors in the dimeric LH1–RC complex.**
**a** Tilted view from the periplasmic side of the membrane with PufX (orange) and proteins-U (magenta) shown by transparent cylinders. Color scheme: BChl *a*, red sticks; spheroidenes (group A), yellow sticks; spheroidenes (group B), orange sticks; BPhe *a*, magenta sticks; UQ-10, purple sticks; nonheme Fe, green sphere. **b** *B*-factor distribution of the carotenoids in the core complex with colors representing the *B*-values indicated in the color bar. BChl *a* and BPhe *a* molecules are colored light gray.

that the carotenoids in monomer-B are generally more rigid than those in monomer A, and the carotenoids in the ring junction are more rigid than carotenoids in other portions of the LH1 ring with similar rigidity to the *cis*-spheroidenes in the RCs. These trends are also consistent with the resolution and *B*-factor distributions over all atoms (Supplementary Fig. 3).

**Junction of the *Rba. sphaeroides* LH1–RC dimer.** Monomers A and B in the dimeric LH1–RC join at the LH1-ring ends near PufX (Fig. 3a). The junction of the two monomers is composed of two PufX and two pairs of LH1 αβ-polypeptides (Fig. 3b). There are no carotenoids present at the joint (between chains AB and ab pairs) of the two rings occupied by PufX (bottom panel in Supplementary Fig. 6). The two PufX polypeptides function as linkers connecting the two monomers (Fig. 4), and due to a largely tilted conformation, the two PufX polypeptides can only interact with each other through their N-terminal domains on the cytoplasmic side of the membrane (Fig. 4a, b). Interactions between charged/polar residues and CH–π interactions primarily occur in the range of Asn8–Phe25. Four tightly bound cardiolipin (CL) and two ubiquinone (UQ) molecules were identified in the junction (Fig. 3 and Supplementary Fig. 7). The head groups of CL-1 and CL-3 interact extensively with the charged/polar residues in the N-terminal domains of the two PufX (Supplementary Fig. 7a), strengthening the connection between the two monomers. CL-2 and CL-4 also interact with PufX and are in close contacts with a different β-polypeptide on the periplasmic side of the membrane (Supplementary Fig. 7b). The head groups of two UQ molecules are in close proximity to PufX and LH1-α polypeptides in the transmembrane region (Supplementary Fig. 7c).

Extensive hydrophobic interactions were observed in the transmembrane region between PufX and the LH1 α-polypeptide positioned at the junction within the same monomer (Fig. 3, Fig. 4a, c). Moreover, several CH-π interactions along with cation–π interactions[6] exist between the two polypeptides. PufX crosses an α-polypeptide through a short, extremely rigid Gly- and Ala-rich stretch (GAGWAGG) (Fig. 4a, Supplementary Fig. 8) as was demonstrated by H/D-exchange NMR[17]. The center of the stretch in PufX points to the Gly21 in the α-polypeptide that is conserved in all *Rhodobacter* species. As a result, the contact interface between the two polypeptides is composed of two crossing grooves formed by the residues with small sidechains (essentially Gly), allowing the closest-possible approach (<4.0 Å) of the two main chains through van der Waals interactions. A similar feature was observed for a PufX/α-polypeptide pair in the monomeric LH1–RC–PufX structure from *Rba. veldkampii*; in this structure, PufX crosses the α-polypeptide through a narrow segment (Gly–Met–Gly) between PufX and the Gly21 in the α-polypeptide (Supplementary Fig. 8c, d)[18]. It is notable that Gly30 of PufX and Gly21 of the α-polypeptide in *Rba. sphaeroides* are conserved in all *Rhodobacter* species. Other, larger residues around the crossing interface in *Rba. sphaeroides* PufX have most of their bulky sidechains pointing outward (Supplementary Fig. 8b).

PufX also interacts with the β-polypeptide that is positioned at the junction and is part of a different monomer (Fig. 3, Fig. 4a, d). Interactions between charged/polar residues in the N-terminal domains (Asn8–His10) and CH–π interactions in the transmembrane domains were observed between these two polypeptides. PufX interacts extensively in the C-terminal domain with the RC L-subunit of the same monomer (Fig. 4a). Such interactions have been described in detail for a monomeric *Rba. sphaeroides* LH1–RC[6] and an LH1–RC–PufX complex from *Rba. veldkampii*[18]. The structural features of PufX in our work are consistent with biochemical analyses, showing that PufX is required for dimerization of the core complex[19–22], and that truncations of N- and C-terminal domains of PufX disrupt formation of a dimeric LH1–RC[21,23].

**Structural comparison of the monomers in the dimer.** Asymmetry was also observed for the LH1 organization of the *Rba. sphaeroides* dimeric LH1–RC structure (Supplementary Fig. 4c). The two monomers revealed different arrangements of the LH1 subunits, especially for the two αβ-pairs near protein-U (Fig. 5a). Superposition of the RC M-subunits between the two monomers showed that the 13–14th αβ-polypeptides in monomer B have gradually shifted inward from their corresponding pairs in monomer A at the end of the LH1 rings, reaching a maximum deviation of approximately 6 Å for the last 14th αβ-pair (chain ID: 07/08) (Fig. 5a). The inward shift of the last two α-polypeptides (chain ID: 05 and 07) in monomer B could cause a conformational disruption of the adjacent protein-U and explain the large disorder observed for protein-U in monomer B. We further tried to elucidate conformational differences between the two independent monomers in the dimer using RELION. Despite applying twice-focused 3D classification[24], each monomer appeared to have only one major conformation that was virtually identical to the corresponding structure after global refinement (Supplementary Fig. 5 and Methods). Therefore, we refer to the global refined structures of the monomers in the dimer as representatives.

The structure of monomer A in the dimeric LH1–RC well overlaps with that of a monomeric *Rba. sphaeroides* LH1–RC[6] from the same wild-type strain of *Rba. sphaeroides* for all protein and pigment components (Fig. 5b). The two copies of protein-U in the dimeric core complex exhibited significantly different

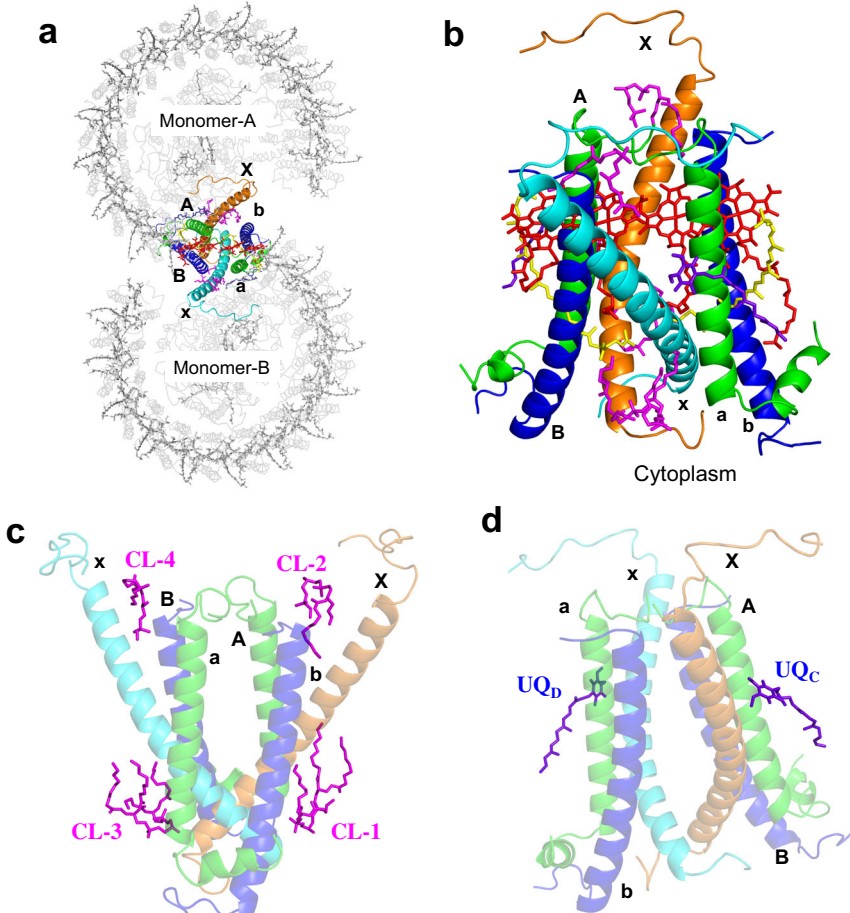

**Fig. 3 Junction of the *Rba. sphaeroides* LH1–RC dimer. a** Top view from the periplasmic side of the membrane with the polypeptides and cofactors in the dimer junction shown in colors and by their chain ID: A, B, and X denote the α- (green), β- (blue), and PufX (orange) polypeptides in monomer A, respectively; a, b, and x denote the α- (green), β- (blue), and PufX (cyan) polypeptides in monomer B, respectively. BChl *a* and spheroidenes are shown by red and yellow sticks, respectively. Cardiolipin (CL) and UQ molecules are shown by magenta and purple sticks, respectively. **b** Side view of the expanded dimer junction with the same color scheme as in (**a**). **c** Four CL molecules (magenta sticks) in the dimer junction. **d** Two UQ molecules (purple sticks) in the dimer junction.

degrees of disorder (Fig. 5c, d). The protein-U in monomer B could only be traced from fragmented densities for 13 amino acids near the cytoplasmic side (Fig. 5d), while other portions were highly disordered. This is likely due to the anticlockwise rotation of monomer A toward the LH1-ring edge of monomer B where protein-U in monomer B is located (Fig. 1a). This rotation would naturally push the last two αβ-subunits in monomer B toward the nearby protein-U (Fig. 5a), and as a result, disrupt its conformation.

**Structure of the *Rba. sphaeroides* LH1–RC ΔU monomer.** In order to further explore structure–function relationships of protein-U, we also determined the cryo-EM structure of a monomeric LH1–RC from a protein-U deletion strain of *Rba. sphaeroides* IL106 (strain ΔU) at 2.63 Å resolution (Supplementary Fig. 9). The ΔU LH1–RC monomer revealed a half-moon-shaped structure containing 11 α- and 10 β-polypeptides (Fig. 6). Three αβ-pairs near where protein-U resides in the wild-type LH1–RC monomer as well as the β-polypeptide paring with the 11th α-polypeptide were not identified in the density map (Fig. 6c). In addition, a group-B spheroidene between the 9th and 10th β-polypeptides was also not observed (Fig. 6b). By contrast, all other proteins and pigments in the ΔU LH1–RC were present and had the same structures as those in the wild-type monomeric LH1–RC[6]. This result clearly indicates that protein-U plays a

critical role in controlling the number of αβ-subunits in an LH1 monomer and that a structurally complete LH1 ring is a pre-requisite for both LH1–RC dimer formation and its subsequent stabilization, in firm agreement with biochemical analyses[6].

**Discussion**

At the resolution of 2.75 Å achieved in our work, the dimeric structure of the native *Rba. sphaeroides* LH1–RC core complex revealed several asymmetric characteristics. These include the arrangement of the two constituting monomers, the structural integrity of protein-U, the organization of LH1, and the rigidity of protein and pigment components in the dimer. These stand in contrast to a twofold symmetric dimer structure reported for the core complex from LH2-deficient *Rba. sphaeroides* strain DBCΩG[7]. The asymmetric features observed in our work may well be related to each other and may have their origins in the bent configuration of the dimer (Fig. 1b) that is likely imposed by the topology of ICM that houses the photocomplexes of purple bacteria.

ICM is formed by invagination of cytoplasmic membrane (CM) with an inverted curvature relative to that of the CM and is known to accommodate major photosynthetic pigment-membrane protein machineries[25]. The ICMs in wild-type *Rba. sphaeroides* cells were demonstrated to have a spherical shape[26,27]. Upon isolation of ICM (referred to as

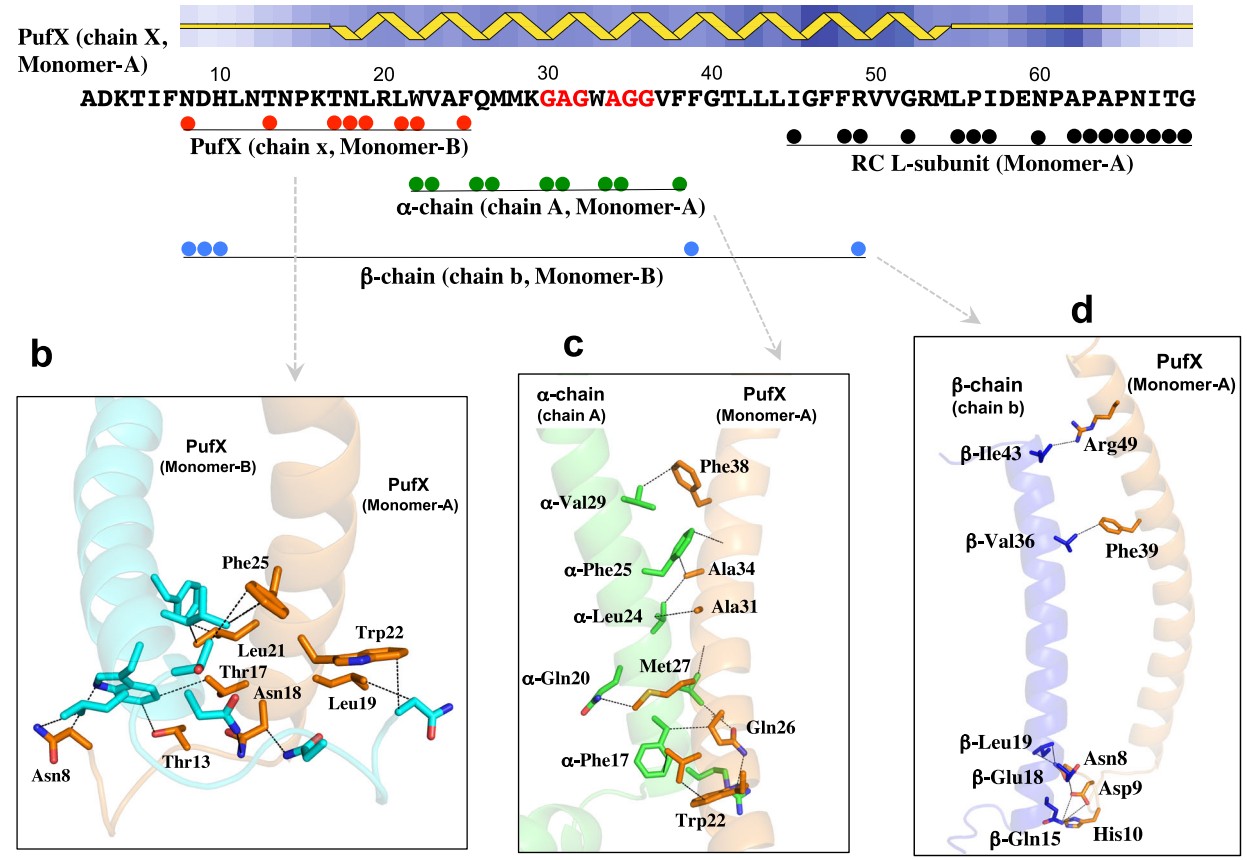

**Fig. 4 Interaction mapping of PufX in monomer A with its partner proteins. a** Primary sequence and secondary structure of the expressed PufX with the color spheres representing its amino acid residues that are in close contacts (<4.0 Å) with partner polypeptides as indicated. **b** Interactions between the two PufX polypeptides in their N-terminal regions with the dashed lines representing typical close contacts between the two polypeptides. **c** Close contacts (<4.0 Å) between PufX and an α-polypeptide of the same monomer in their transmembrane regions. The crossing portion in PufX corresponds to a short, Gly- and Ala-rich stretch as indicated by red fonts in the sequence of (**a**). **d** Close contacts between PufX and a β-polypeptide of a different monomer in their N-terminal and transmembrane regions.

chromatophores), the periplasmic surface of photosynthetic membranes is positioned on the curved-in side of the chromatophores (inside of the spherical vesicles)[28], consistent with the curvature of a bent configuration of the LH1–RC dimer. The uneven rigidities over a dimension of approximately 190 Å of the native dimeric LH1–RC (Fig. 2b, Supplementary Fig. 3) likely reflect the inhomogeneity and fluidity of the photosynthetic membranes[29], and the carotenoid molecules distributed in the dimer can be used as indicators of rigidity for the entire core complex.

In contrast to the structure of chromatophores from wild-type *Rba. sphaeroides*, ICM from a mutant strain lacking LH2 exhibited a flattened tubular morphology[30]. These differences may be the result of one (or both) of two possibilities. First, the ICM from cells lacking LH2 may have a significantly different lipid composition than that of ICM from wild-type cells. Differences in the lipid composition, coupled with the total absence of LH2, likely change native ICM structure in dramatic ways. Second, a new protein has been reported from dimeric LH1–RC complexes prepared from the same LH2-deficient strain, a protein not observed in the dimeric complexes we prepared from wild-type cells. This protein, referred to as protein Z[7], is also absent from monomeric LH1–RC complexes from the LH2-deficient mutant[31]; because of this specific association, protein Z was proposed to play a role in the formation of LH1-RC dimers[7]. On the contrary, our structure of the native complex yielded no

evidence for a Z-like protein, and therefore it should not be necessary for formation of dimeric complexes in wild-type cells.

PufX in the *Rba. sphaeroides* LH1–RC complex plays a critical role in formation of the dimeric core complex, and our cryo-EM structure of PufX in a dimer provides a foundation for examining its structural and functional roles that have been investigated only biochemically thus far. Deletion of the gene encoding *Rba. sphaeroides* PufX results in a mutant strain that can only synthesize a monomeric form of the core complex[19–22]. Moreover, truncations of either the N- or C-terminal domain[21,23] or point mutations of certain PufX residues[21,22] also disabled dimer formation. These results now can be explained by an interaction map of PufX that shows that one LH1–RC monomer interacts with the PufX and a β-polypeptide of a second monomer primarily through their N-terminal domains (Asn8–Phe25 in PufX, Fig. 4), thus connecting the two monomers. This structure is consistent with mutagenic studies that show that removal of 12 or more residues from the N-terminus of PufX results in the complete loss of dimeric complexes even though the truncated PufX polypeptides are expressed and some of the mutants still retained the ability to grow photosynthetically[21,23]. Four CL molecules interacting with PufX and β-polypeptides in the junction also stabilize the dimeric form (Fig. 3c). However, these PufX-mediated intermonomer interactions can be disrupted in vitro with relatively high concentrations of the detergent *n*-octyl-β-glucopyranoside[19], resulting exclusively in monomeric forms.

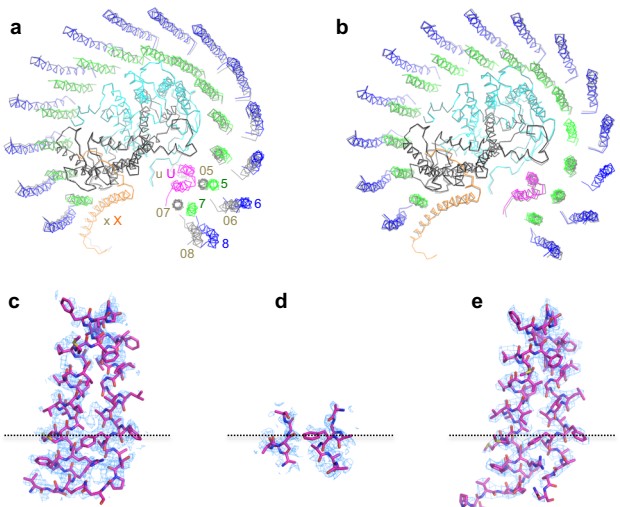

**Fig. 5 Structural comparison of the monomers in the dimer. a** Top view of superposition of the Cα carbons of the RC M-subunits between monomer A (colored) and monomer B (gray) shown by ribbon representations and viewed from the periplasmic side. Color scheme for the proteins in monomer A is the same as in Fig. 1. PufX (X, x) and protein-U (U, u) are indicated by their chain ID (X and U in monomer A, x and u in monomer B). The RC H-subunits are omitted for clarity. Two αβ-subunits in monomer B indicated by their chain ID 05, 06, 07, and 08 are largely deviated from the corresponding polypeptides in monomer A (chain ID 5, 6, 7 and 8). **b** Similar superposition between monomer A (colored) and a monomeric *Rba. sphaeroides* LH1–RC (gray, PDB: 7F0L) reported previously. **c–e** Structures of protein-U with their density maps at a contour level of 4σ in monomer A (**c**), monomer B (**d**), and the monomeric core complex (**e**). Dotted lines indicate the position of Phe46 of protein-U. Most parts of protein-U in monomer B are disordered.

*Rba. sphaeroides* PufX interacts with the RC L-subunit in a monomer over a wide range of its C-terminal domain (Ile45–Gly69, Fig. 4a)[6]. Point mutations of PufX Arg49, Gly52, and Arg53 to Leu resulted in loss of the dimeric LH1–RC; nevertheless, these mutant strains still grow photosynthetically at rates similar to that of wild-type[22]. The Arg49 and Gly52 in PufX are in close contacts with the RC L-subunit near the periplasmic surface as shown in Fig. 4a. By contrast, deletions of 7 or more residues from the C-terminus that are encoded in the *pufX* gene result in the absence of the truncated PufX in core complexes and assembly of only monomeric forms[23]; such results point to a major role for the C-terminus of PufX in its assembly in the monomeric complex. Compared with N- and C-terminal domains, hydrophobic interactions in the transmembrane regions of PufX and an α-polypeptide of the same monomer is likely less critical for dimer formation. For example, mutations of several Gly residues to Leu in the central stretch of PufX did not decrease the propensity of the core complex to assemble in a dimeric form[20]. Overall, the interaction features of PufX with its partner proteins revealed in our study are in good agreement with biochemical analyses and support the conclusion that the N-terminal domain of PufX is directly responsible for dimerization, while the C-terminal domain mainly participates in assembly of the core complex and along with the transmembrane region stabilizes the dimeric structure.

Another important finding of our work concerns the role of protein-U. Although protein-U has no inter-monomeric interactions in the *Rba. sphaeroides* dimeric LH1–RC, it clearly plays a role in regulating LH1-ring size. Appropriate ring size is crucial for assembly and stabilization of the dimeric complex in order for it to function in a curved-membrane topology. This is consistent with biochemical analyses that show that strain ΔU can still produce a dimeric complex but at much reduced levels compared with the wild type[6]. A recent structure of the LH1–RC from *Rba.*

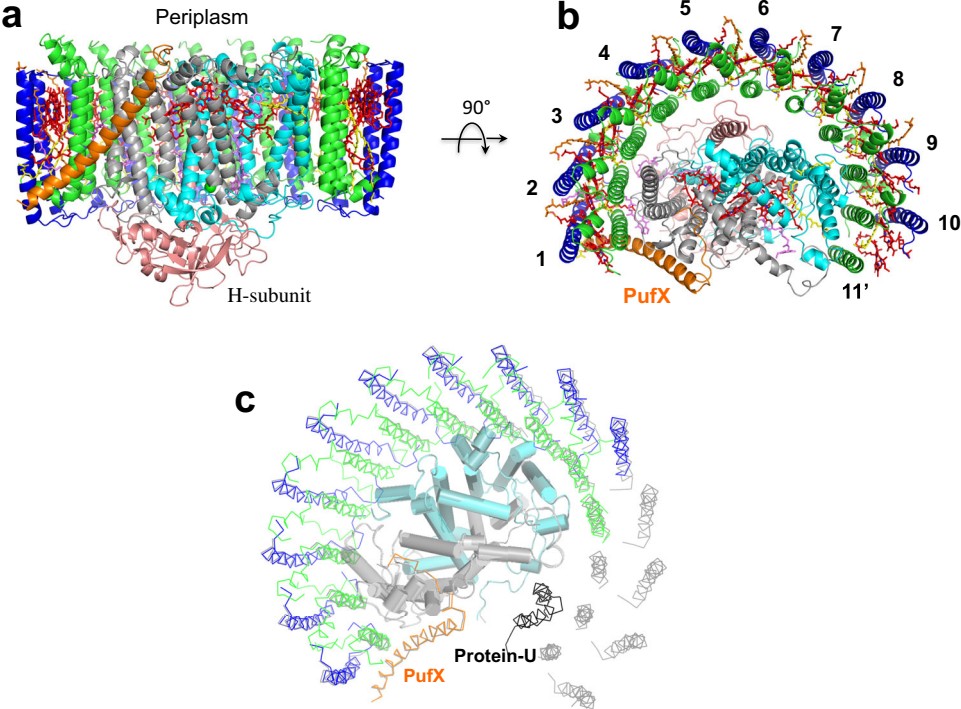

**Fig. 6 Structure of the *Rba. sphaeroides* LH1–RC ΔU monomer. a** Side view of the complex with PufX (orange) in the front. Color scheme is the same as in Fig. 1. **b** Top view from periplasmic side of the membrane with numbering for the LH1 polypeptides. **c** Overlap view of the ΔU monomer (colored) and wild-type monomeric *Rba. sphaeroides* LH1–RC (gray, PDB: 7F0L) by superposition of Cα carbons of the RC M-subunits. Protein-U as it exists in the wild-type monomer is shown by a black ribbon.

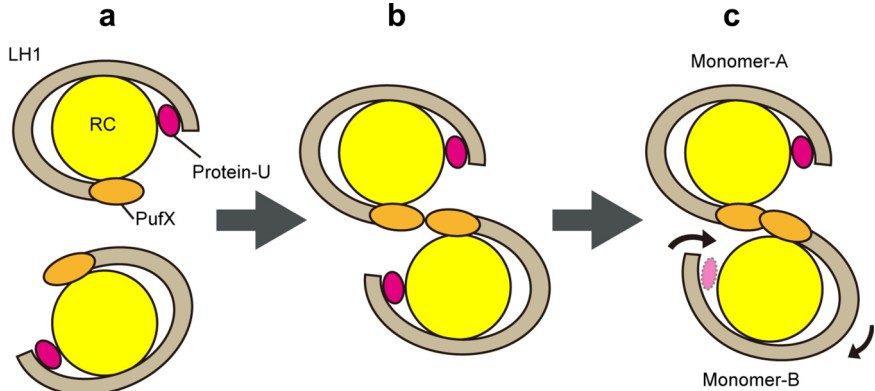

**Fig. 7 Schematic docking model for the dimerization of LH1–RC complex.** Cartoon representation of the complexes viewed from the periplasmic side.
**a** Two C-shaped LH1–RC monomers before dimerization. **b** Dimerization of the monomers at an early stage with a symmetric S-shaped conformation.
**c** Formation of an asymmetric junction is completed with breaking symmetry. Due to a wind-up movement of monomer B, most parts of protein-U are disordered and the LH1-ring opening is narrowed in monomer B. Thus, the sizes of LH1-ring opening are different to enhance asymmetric features. Because monomeric ΔU LH1–RC complex lacks protein-U and only has a half-LH1 ring, it could not form a stable dimer. The color codes are RC: yellow, LH1 ring: gray, PufX: orange, Protein-U: magenta.

*veldkampii* showed that its LH1 is composed of 15 αβ-subunits (Supplementary Fig. 10d)[18], one subunit more than in the *Rba. sphaeroides* LH1. *Rba. veldkampii* only synthesizes the monomeric form of LH1–RC and lacks protein-U in its core complex, further supporting our conclusion that protein-U in *Rba. sphaeroides* regulates the number of αβ-subunits in the complex, thereby controlling the size of the LH1-ring opening.

Based on the structures determined in our work and that of others, we propose a model for assembly of the dimeric LH1–RC complex in native membranes of wild-type *Rba. sphaeroides* (Fig. 7). In our model, the process begins when two structurally complete C-shaped LH1–RC monomers collide and link at their PufX sites. Protein-U does not directly contribute at this stage but ensures the correct LH1-ring size required for initial dimerization. The dimer retains a symmetric conformation (Fig. 7b) at this point, but is likely in a transient state because of the instability associated with remaining in a curved-membrane environment composed of a heterogeneous mixture of lipids. To accommodate the curved topology, the symmetric dimer then undergoes a spontaneous adjustment in which the LH1 ring of one monomer constricts (forming a smaller gap), and this triggers the disorder of its protein-U (Fig. 7c). Such an event is supported by inspection of the asymmetric distortion of the lipid/micelle belts surrounding the dimeric LH1–RC (Supplementary Fig. 11). Monomer A has a larger LH1 ring opening than monomer B and its lipid/micelle belt is discontinued in the gap region (Supplementary Fig. 11b). By contrast, the narrow gap of monomer B results in a continuous lipid/micelle belt over the junction region (Supplementary Fig. 11d) where four tightly bound CL molecules are present (Fig. 3c). These differences between the two monomers still remained even after applying focused 3D classification[24], confirming the local structural features independent of global alignment. It therefore follows that monomer B should have the most stable conformation of the two monomers due to more extensive interactions with the surrounding lipids, and as a result, an asymmetric rather than a symmetric dimer forms. In our model, the asymmetric dimer observed in membranes of wild-type *Rba. sphaeroides* thus represents the final mature and functional form of the LH1–RC complex.

## Methods
### Preparation and characterization of the native dimeric and ΔU monomeric LH1-RC complexes.
*Rba. sphaeroides* f. sp. *denitrificans* (strain IL106) cells[32] were cultivated phototrophically (anoxic/light) at room temperature for 7 days under

incandescent light (60 W). Preparation of the native LH1–RC was conducted by solubilizing chromatophores ($OD_{870\text{-nm}} = 40$) with 1.0% w/v *n*-dodecyl-β-D-maltopyranoside (DDM) in 20 mM Tris-HCl (pH 8.0) buffer for 60 min at room temperature, followed by differential centrifugation. The supernatant was loaded onto a DEAE column (Toyopearl 650 S, TOSOH) equilibrated at 4 °C with 20 mM Tris-HCl buffer (pH 8.0) containing 0.1% w/v DDM. The fractions were eluted in an order of LH2, monomeric LH1–RC, and dimeric LH1–RC by a linear gradient of NaCl from 0 mM to 400 mM. The dimer-rich LH1–RC fractions were collected and further purified by gel-filtration chromatography (TSKgel G4000SW, 7.5 mm I.D. × 60 cm, TOSOH) with 20 mM Tris-HCl buffer (pH 7.5) containing 200 mM NaCl and 0.05% w/v DDM at a flow rate of 1 mL/min at room temperature (Supplementary Fig. 1b). Monomeric LH1–RC lacking protein-U was solubilized from ICM of *Rba. sphaeroides* IL106 strain ΔU[6] using 1.0% w/v DDM, followed by sucrose gradient-density centrifugation with five-stepwise sucrose concentrations (10, 17.5, 25, 32.5 and 40% w/v) in 20 mM Tris-HCl buffer (pH 8.0) containing 0.05% w/v DDM at 4 °C and 150,000 × g for 6 hours. Both native dimeric LH1–RC and monomeric ΔU LH1–RC were concentrated for absorption measurement and assessed by negative-stain EM using a JEM-1010 instrument (JEOL).

**Cryo-EM data collection.** Proteins for cryo-EM were concentrated to ~5 and 3.4 mg/ml for the native dimer and the ΔU mutant, respectively. Three microliters of the protein solution were applied on a glow-discharged holey carbon grids (200 mesh Quantifoil R2/2 molybdenum), which had been treated with $H_2$ and $O_2$ mixtures in a Solarus plasma cleaner (Gatan, Pleasanton, USA) for 30 s and then blotted, and plunged into liquid ethane at –182 °C using an EM GP2 plunger (Leica, Microsystems, Vienna, Austria). The applied parameters were a blotting time of 4 s at 80% humidity and 4 °C. Data were collected on a Titan Krios (Thermo Fisher Scientific, Hillsboro, USA) electron microscope at 300 kV equipped with a Falcon 3 camera (Thermo Fisher Scientific) (Supplementary Fig. 2). Movies were recorded using EPU software (Thermo Fisher Scientific) at a nominal magnification of 96 k in counting mode and a pixel size of 0.82 Å at the specimen level with a dose rate of 1.03 e- per physical pixel per second, corresponding to 1.25 e- per Å² per second at the specimen level. The exposure time was 32.0 s, resulting in an accumulated dose of 40 e- per Å². Each movie includes 40 fractioned frames.

**Image processing.** All of the stacked frames were subjected to motion correction with MotionCor2[33]. Defocus was estimated using CTFFIND4[34]. All of the picked particles using the crYOLO[35] were further analyzed with RELION 3.1[36], and selected by 2D classification (Supplementary Fig. 2, Supplementary Fig. 9, and Supplementary Table 1). Initial 3D model was generated in RELION, and the particles were divided into four classes by 3D classification, resulting in only one good class. The 3D autorefinement without any imposed symmetry (C1) produced a map at 2.88 Å and 2.70 Å resolutions for the native dimer and ΔU monomer, respectively, after contrast-transfer function refinement, Bayesian polishing, masking, and post-processing. Then particle projections were further subjected to subtraction of the detergent-micelle density followed by 3D autorefinement to yield the final map with 2.75 Å and 2.63 Å resolutions for the native dimer and ΔU monomer, respectively, according to the gold-standard Fourier shell correlation (FSC) using a criterion of 0.143 (Supplementary Fig. 2, Supplementary Fig. 9 for the native dimer and ΔU monomer, respectively)[37]. The local resolution maps were calculated on RESMAP[38].

The procedure of focused classification of individual monomers in an oligomer using mask[24] in RELION has been described[39,40]. In our focused 3D classification,

four classes of 3D maps for monomer-A and -B were computed independently from the corresponding position of the final particle images for dimer reconstruction, until FSCs were converged, each of which showed unique conformations of the native monomer (Supplementary Fig. 5a). After checking these maps manually, only a single class in each form showed well-resolved density, such as secondary structures. Because other maps were featureless (and thus unsuitable for model building), we did not use the particles derived from these classes. Finally, we performed another round of focused 3D classification with tau_fudge = 20. Resolutions of the maps for monomer A and -B were estimated to be 3.7 Å and 3.6 Å (Supplementary Fig. 5a), respectively.

**Model building and refinement of the native dimeric and ΔU monomeric LH1–RC complexes.** The atomic model of the monomeric LH1–RC complex of *Rba. sphaeroides* (PDB: 7F0L) was fitted to the cryo-EM native dimer and ΔU monomer maps, respectively, using Chimera[41]. Real-space refinement for the peptides and cofactors was performed using COOT[42]. The manually modified model was refined in real space on PHENIX[43], and the COOT/PHENIX refinement was iterated until the refinements converged. Finally, the statistics calculated using MolProbity[44] were checked. Figures were drawn with the Pymol Molecular Graphic System (Schrödinger)[45] and UCSF Chimera[41].

**Reporting summary**. Further information on research design is available in the Nature Research Reporting Summary linked to this article.

## Data availability

The map and model generated in this study have been deposited in the EMDB and PDB with accession codes: EMD-32192 and PDB-7VY2 for the native LH1–RC dimer and EMD-32193 and PDB-7VY3 for the ΔU LH1–RC monomer.

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

## Acknowledgements

This research was partially supported by Platform Project for Supporting Drug Discovery and Life Science Research (Basis for Supporting Innovative Drug Discovery and Life Science Research (BINDS)) from AMED under Grant Numbers JP20am0101118 (support number 1758) and JP20am0101116 (support number 1878), 17am0101116j0001, 18am0101116j0002 and 19am0101116j0003. R.K., M.H. A.T. and B.M.H. acknowledge the generous support of the Okinawa Institute of Science and Technology and the Japanese Cabinet Office. B.M.H, R.K, and M.H. thank the members of the Scientific Computing & Data Analysis Section of OIST for their continuous support. L.-J.Y. acknowledges support of the National Key R&D Program of China (No. 2019YFA0904600). This work was supported in part by JSPS KAKENHI Grant Numbers JP16H04174, JP18H05153, JP20H05086, and JP20H02856.

## Author contributions

Z.-Y.W.-O., and K.T. designed the work, K.V.P.N. provided materials, K.T., R. Kanno, R. Kikuchi, S.K., M.H., and A. T. performed the experiments, K.T., R. Kanno, L.-J.Y., Y.K., M.T.M., A.M., B.M.H., and Z.-Y.W.-O. analyzed data, and Z.-Y.W.-O., K.T., and M.T.M. wrote the paper.

## Competing interests

The authors declare no competing interests.
