## [Peer Review File · Nature Communications]

Asymmetric Structure of the Native Rhodobacter sphaeroides Dimeric LH1-RC ComplexREVIEWER COMMENTS

Reviewer #1 (Remarks to the Author):

The manuscript by Tani and coworkers describes two three-dimensional structures of the LH1-RC complex from *R. sphaeroides* as determined using cryo-EM. One complex is from wild type and has a dimeric form while the other is from a DeltaU strain that produces a monomeric form. The EM data appears to have been carefully analyzed resulting in high-quality maps and structures. The results are centered on a few topics. In general, the results are well written and the text flows well. The dimer structure is discussed with an emphasis on inequivalent features for the two proteins of the dimeric form. Two other points of emphasis are the role of PufX and protein-U. Also, the structure of the monomeric form is presented.

Despite the high quality of the work, I recommend against publication in Nature Communications. There has been a plethora of very recent manuscripts describing various structures of the LH1-RC complex as determined using cryo-EM. I find that this manuscript largely duplicates those earlier publications. The roles of PufX and protein-U have been presented in refs 7 and 18, the structures of monomeric, dimeric, and open complexes are presented in refs 8, the previous publication of the authors, Tani Biochemistry 2021 (not cited), and Swainsbury Science Advances 2021 (not cited). Thus, the broad, basic points of this manuscript have already been presented in other publications. The published structures are from different bacteria, but the general features of the complexes are well conserved with no striking differences. This manuscript does contain valuable information, in particular a detailed discussion the differences between the pseudo-related proteins of the dimer form, but this discussion is of a detailed nature that is more suitable for presentation in a more specialized journal.

Reviewer #2 (Remarks to the Author):

The manuscript by Tani et al describes the structure of the LH1-RC-PufX dimer from *R. Sphaeroides*. This structure follows a recently published structure from the same bacteria and determined to a similar (slightly lower) resolution. Still, the quality of the current manuscript, the added insights obtained from C1 refinement and the structure of the LH1-RC monomer from the delta protein-U strain merits publication in Nature communication in my opinion. In addition to being of interest to the photosynthetic community this study also addresses biological heterogeneity and assembly issues which are of interest to a general scientific audience.

The technical quality of manuscript is very good, the authors conclusions are largely supported by their data and the data itself is presented very well. My biggest issue with the current manuscript is the quality of the map at and around protein-U. The authors acknowledge this issue, but it still remains that in their provided sharpened maps protein-U is barely seen over background levels, the same can be said for at least the terminal alpha subunit in their monomer structure. This issue should be resolved by either presenting the unsharpen maps for these features (assuming they support the assignment better) or any other method. The authors used C1 refinement to obtain their map, but they do not mention any focused refinement or classification done on individual monomers which can potentially resolve this issue as well.

Supplementary Fig. 8 – Missing phase randomized FSC curve.

Response to reviewers:

Reviewer #1

Reviewer #1's comments

The manuscript by Tani and coworkers describes two three-dimensional structures of the LH1-RC complex from *R. sphaeroides* as determined using cryo-EM. One complex is from wild type and has a dimeric form while the other is from a DeltaU strain that produces a monomeric form. The EM data appears to have been carefully analyzed resulting in high-quality maps and structures. The results are centered on a few topics. In general, the results are well written and the text flows well. The dimer structure is discussed with an emphasis on inequivalent features for the two proteins of the dimeric form. Two other points of emphasis are the role of PufX and protein-U. Also, the structure of the monomeric form is presented.

Despite the high quality of the work, I recommend against publication in Nature Communications. There has been a plethora of very recent manuscripts describing various structures of the LH1-RC complex as determined using cryo-EM. I find that this manuscript largely duplicates those earlier publications. The roles of PufX and protein-U have been presented in refs 7 and 18, the structures of monomeric, dimeric, and open complexes are presented in refs 8, the previous publication of the authors, Tani Biochemistry 2021 (not cited), and Swainsbury Science Advances 2021 (not cited). Thus, the broad, basic points of this manuscript have already been presented in other publications. The published structures are from different bacteria, but the general features of the complexes are well conserved with no striking differences. This manuscript does contain valuable information, in particular a detailed discussion the differences between the pseudo-related proteins of the dimer form, but this discussion is of a detailed nature that is more suitable for presentation in a more specialized journal.

Our response:

We thank the reviewer for taking time in thoroughly reading our manuscript and appreciate the reviewer's positive assessment of the quality of our work. We would like to emphasize that our work is not "just another complex", but a complex with many new significant features that have allowed us to answer important and longstanding questions that were not obvious from previous research. Our work, which reached a quite different conclusion concerning *Rba. sphaeroides* dimeric LH1-RC structure from that reported recently by another group, was performed on the most widely researched wild-type strain of this organism and forms a natural complement to our recent report of a previously unrecognized integral membrane protein (protein-U) in the LH1-RC complex from the same wild-type strain. We thus feel that our results are crucial for understanding the structure and function of this complex as it exists in the *native* membrane environment.

- We also appreciate the reviewer's insightful comments on the difference between the pseudo-related proteins of the dimer formation. This reviewer well understands our conclusion that dimer formation breaks the symmetry, but our incomplete description caused a misunderstanding that the result may not be of interest to a general scientific audience. However, as point out by reviewer #2, the structural heterogeneity and assembly of the dimer are important issues that need to be addressed in more details. For this purpose, we have taken a significant amount of time and computational effort to perform focused 3D classifications on the local maps. Despite applying twice focused 3D classification, each monomer appeared to have only one major conformation that is virtually identical to the corresponding structure after global refinement. Therefore, we refer to the globally refined structures of the monomers in the dimer as representatives, and tried to show to a general audience that cryo-EM structure determination needs to be careful and our structural analyses in this work are unbiased. We

have added Supplementary Fig. 5, three references and descriptions in main text and Methods to address this issue (red fonts).

- In addition to our dimeric structure, our paper includes the structure of the LH1-RC from a protein-U knockout mutant of *Rba. sphaeroides* (strain ΔU), clarifying the structural and functional roles of this newly discovered protein. In combination, these findings allowed us to build an assembly model for dimerization of the LH1-RC complex in native membranes.
- Finally, we emphasize that our study lays the necessary foundation for further advancing studies of photosynthetic light reactions in the most widely used model phototrophic bacterium and should thus appeal to a broad scientific audience, particularly those whose interests include photosynthesis and bioenergetics, structural biology, and the diversity of photosynthetic life on Earth.
- We tried to select only closely relevant literature for the reference list in our manuscript in order to make the content more focused and concise rather than a general review.

Reviewer #2

Reviewer #2's comments

The manuscript by Tani et al describes the structure of the LH1-RC-PufX dimer from *R. Sphaeroides*. This structure follows a recently published structure from the same bacteria and determined to a similar (slightly lower) resolution. Still, the quality of the current manuscript, the added insights obtained from C1 refinement and the structure of the LH1-RC monomer from the delta protein-U strain merits publication in Nature communication in my opinion. In addition to being of interest to the photosynthetic community this study also addresses biological heterogeneity and assembly issues which are of interest to a general scientific audience.

The technical quality of manuscript is very good, the authors conclusions are largely supported by their data and the data itself is presented very well. My biggest issue with the current manuscript is the quality of the map at and around protein-U. The authors acknowledge this issue, but it still remains that in their provided sharpened maps protein-U is barely seen over background levels, the same can be said for at least the terminal alpha subunit in their monomer structure. This issue should be resolved by either presenting the unsharpen maps for these features (assuming they support the assignment better) or any other method. The authors used C1 refinement to obtain their map, but they do not mention any focused refinement or classification done on individual monomers which can potentially resolve this issue as well.

Supplementary Fig. 8 – Missing phase randomized FSC curve.

Our response:

We appreciate the reviewer's positive assessment of our work. We have taken a significant amount of time and computational effort (CPU time-consuming calculations) to perform focused 3D classifications on the maps as requested by the reviewer. Despite applying twice focused 3D classification, each monomer appeared to have only one major conformation that is virtually identical to the corresponding structure after global refinement. Therefore, we refer to the globally refined structures of the monomers in the dimer as representatives. We have added Supplementary Fig. 5, three references and descriptions in main text and Methods to address this issue (red fonts).

- We have emphasized the unsharpened density corresponding to the protein-U as magenta solid surface in Supplementary Fig. 5b.
- Supplementary Fig. 4a and 4b are still kept to show the unsharpened map, so it would be easy to distinguish the protein-U in each independent monomer. We have modified the phrase from "cryo-EM density map" to "cryo-EM unsharpened density map" in the legend of Supplementary Fig. 4.

- We have added phase randomized (orange), both masked (blue) and unmasked (black) FSC curves in Supplementary Fig. 9d (Supplementary Fig. 8d in old version).
- Figure numbers of Supplementary Fig. 5–10 in old version have been changed to Supplementary Fig. 6–11 in the revised manuscript.

REVIEWERS' COMMENTS

Reviewer #2 (Remarks to the Author):

all my comments were addressed.

Response to the reviewer:

Reviewer #2

Reviewer #2's comment

all my comments were addressed.

Our response:

We thank the reviewer for taking time in thoroughly reading our manuscript and appreciate the reviewer's positive assessment of our work.